# Dogs as Sentinels for Flavivirus Exposure in Urban, Peri-Urban and Rural Hanoi, Vietnam

**DOI:** 10.3390/v13030507

**Published:** 2021-03-19

**Authors:** Long Pham-Thanh, Thang Nguyen-Tien, Ulf Magnusson, Vuong Bui-Nghia, Anh Bui-Ngoc, Duy Le-Thanh, Åke Lundkvist, Minh Can-Xuan, Thuy Nguyen-Thi Thu, Hau Vu-Thi Bich, Hu Suk Lee, Hung Nguyen-Viet, Johanna Lindahl

**Affiliations:** 1International Livestock Research Institute (ILRI), Hanoi 10000, Vietnam; Thang.T.Nguyen@cgiar.org (T.N.-T.); H.S.Lee@cgiar.org (H.S.L.); H.Nguyen@cgiar.org (H.N.-V.); J.Lindahl@cgiar.org (J.L.); 2Department of Medical Biochemistry and Microbiology, Uppsala University, 75123 Uppsala, Sweden; ake.lundkvist@imbim.uu.se; 3Department of Animal Health, Ministry of Agriculture and Rural Development, Hanoi 10000, Vietnam; 4Department of Clinical Sciences, Swedish University of Agricultural Sciences, 75123 Uppsala, Sweden; Ulf.Magnusson@slu.se; 5National Institute for Veterinary Research, Hanoi 10000, Vietnam; buinghiavuong@gmail.com (V.B.-N.); buingocanh_1980@yahoo.com (A.B.-N.); lethanhduytb74@gmail.com (D.L.-T.); 6Hanoi Sub-Department of Livestock Production and Animal Health, Hanoi 10000, Vietnam; minhcx@gmail.com; 7National Institute for Hygiene and Epidemiology, Hanoi 10000, Vietnam; ticun_2002@yahoo.com (T.N.-T.T.); vtbh@nihe.org.vn (H.V.-T.B.)

**Keywords:** dogs, mosquito-borne flavivirus, seroprevalence, Hanoi, Vietnam

## Abstract

Diseases caused by flaviviruses, including dengue fever and Japanese encephalitis, are major health problems in Vietnam. This cross-sectional study explored the feasibility of domestic dogs as sentinels to better understand risks of mosquito-borne diseases in Hanoi city. A total of 475 dogs serum samples from 221 households in six districts of Hanoi were analyzed by a competitive enzyme-linked immunosorbent assay (cELISA) for antibodies to the pr-E protein of West Nile virus and other flaviviruses due to cross-reactivity. The overall flavivirus seroprevalence in the dog population was 70.7% (95% CI = 66.4–74.8%). At the animal level, significant associations between seropositive dogs and district location, age, breed and keeping practice were determined. At the household level, the major risk factors were rural and peri-urban locations, presence of pigs, coil burning and households without mosquito-borne disease experience (*p* < 0.05). Mosquito control by using larvicides or electric traps could lower seropositivity, but other measures did not contribute to significant risk mitigation of flavivirus exposure in dogs. These results will support better control of mosquito-borne diseases in Hanoi, and they indicate that dogs can be used as sentinels for flavivirus exposure.

## 1. Introduction

Viruses within the genus Flavivirus of the family Flaviviridae are responsible for a number of vector-borne diseases in humans, such as dengue, Japanese encephalitis (JE), Zika, yellow fever, West Nile (WN) and many others worldwide [1]. 

In nature, mosquito-borne flaviviruses circulate between arthropod vectors, generally *Aedes* spp. mosquitoes for dengue virus (DENV), Zika virus (ZIKV) and yellow fever virus (YFV), and *Culex* spp. mosquitoes for Japanese encephalitis virus (JEV) and West Nile virus (WNV), and vertebrate hosts [2]. Mosquitoes acquire flaviviruses mainly through horizontal transmission by taking a bloodmeal from a viremic animal, or possibly through vertical transmission from mother to offspring [3], while vertebrate hosts become infected by the probing process of blood feeding of an infected mosquito vector [4]. 

The rapidly urbanizing Hanoi, the capital of Vietnam, has a high density of people and different domestic animals [5]. In 2018, there were 7.9 million people, 1.8 million pigs, 136 thousand cattle, 23.5 thousand buffaloes, 11.5 thousand goats, 0.4 thousand horses, 31.5 million poultry and 450.3 thousand dogs in Hanoi [6]. People in the city are exposed to flaviviruses, mainly DENV and JEV [7]. 

In Hanoi, there are indigenous and exotic breeds of dogs, as well as crossbreeds. They are important as companion pets, for guarding property or as a human food source. Dogs are the closest animals to human dwellings, and they could be exposed to vector-borne pathogens to the same extent as their owners. Due to very low level of viremia, dogs do not usually show any clinical signs of flaviviral infections, nor transmit the disease to humans, but flavivirus seroprevalences in dog populations may be valuable as sentinels to evaluate risk factors for humans [8,9]. The objective of this study was to assess the association between risk factors at the household level and flavivirus exposure in the dog population in Hanoi city.

## 2. Materials and Methods

### 2.1. Study Design

A total of six districts including two more rural, where large populations of livestock are kept (with more than 1000 large ruminants, 15,000 pigs and 150,000 poultry per district), two peri-urban (less than 1000 large ruminants, 15,000 pigs and 150,000 poultry per district) and two urban districts with no livestock keeping in Hanoi were purposively selected to represent a gradient of livestock keeping.

Sample size was calculated as 475 dogs with 50% of the assumed true seroprevalence due to no previous data available for flavivirus prevalence in the dog population of Hanoi city, 5% desired precision, a 95% confidence level and a test assumed with 95% sensitivity and 95% specificity [10]. An additional 5% of the sample size was compensated in case of insufficient samples for testing.

The multi-stage cluster sampling strategy was applied in which random selection of 120 global positioning system (GPS) points in six districts was conducted, and within a radius of 2 km from each GPS point, about five households keeping a dog(s) were visited during September and October 2018. Here, the owners were interviewed by a structured questionnaire (Appendix A, Household questionnaire) form that was pre-tested for comprehensibility, of which demographic characteristics of the respondents and their dogs, potential risk factors related to livestock keeping and mosquito prevention practices were included, and dog blood was taken for serology. A house keeping at least a ruminant or a pig or five small animals such as chickens, ducks, geese or rabbits was defined as a livestock-keeping household. 

### 2.2. Sample Collection and Storage

Dog blood sampling was conducted by trained veterinarians of the Hanoi Sub-Department of Livestock Production and Animal Health, using venipuncture of *Vena cephalica* or *V. saphena*. If a household had several dogs, the maximum number of dogs sampled was five. The sample size was calculated from 50% of the expected seroprevalence within a flock with a 95% confidence level, and the same test sensitivity and specificity levels at 95%. The samples were stored in a cool box in the field and transferred to the National Institute for Veterinary Research (NIVR) on the sampling day. Sera were centrifuged and separated immediately and kept at −20 °C condition until tested. A total of 502 dogs from 225 households surrounding 44 GPS points in six districts of Hanoi were sampled (Table 1). 

### 2.3. Laboratory Technique

#### Competitive Enzyme-Linked Immunosorbent Assay (cELISA)

A commercial cELISA kit for detection of IgG antibodies against WNV manufactured by IDvet company (Grabels, France) was employed. In principle, samples to be tested and controls are added to the microwells precoated with the pr-E protein of WNV. However, the protein includes epitopes that are common to all flaviviruses, causing serological cross-reactions; hence, this ELISA kit does not only detect antibodies against WNV but also cross-reactive antibodies induced by other flaviviruses [11]. Anti-pr-E antibodies, if present, form an antigen–antibody complex. An anti-pr-E antibody horseradish peroxidase (HRP) conjugate binds to the remaining free pr-E epitopes, forming an antigen–conjugate–peroxidase complex. The kit is not species-dependent [8,12,13]. The analyses were performed according to the manufacturer’s instructions. Calculation of the S/N percentage (S/N%) was equal to the value of optical density (OD) of the sample divided by the value of OD of the negative control then multiplied by 100. Samples presenting an S/N% less than or equal to 40% were considered positive; higher than 40% and less than or equal to 50% were considered doubtful; higher than 50% were considered negative. 

Each dog serum was tested in duplicate on the same ELISA plate.

### 2.4. Statistical Analysis

A total of twenty-seven serum samples were excluded in the analysis: 16 samples had insufficient volumes and 11 samples showed doubtful results by the cELISA. Four households that had all their samples within the group of the 27 excluded sera were removed. Households were considered positive if at least one dog showed seropositivity. Data obtained from the questionnaires and the laboratory results were recorded in an Excel^®^ spreadsheet and then transferred into STATA/SE 15.0 (StataCorp LLC, College Station, TX, USA) for analysis. Descriptive statistics for the categorical variables for dogs and households displayed by the cELISA results were used. A chi-square test was used to evaluate the association of the explanatory variables in the univariable analyses. At the household level, all independent variables were compared to assess the correlation among variables. A stepwise selection of variables based on univariable analyses with a cutoff value of 0.25 [14] and correlation measurements was applied in logistic regression models. Variables changing more than 25% from the coefficients of other variables were classified as confounding factors and they were forced into the model if the affected variables were significant. The likelihood ratio test was performed to build a parsimonious model, and the Hosmer–Lemeshow goodness-of-fit test was used for model fitness [15]. A *p*-value <0.05 was considered statistically significant.

### 2.5. Ethics

Ethical approval was obtained from the Ethical Review Board for Biomedical Research of Hanoi University of Public Health (Number 406/2018/YTCC-HD3, approved on 6 August 2018). The purpose of this study was explained to dog owners by veterinary officers of the Hanoi Sub-Department of Livestock Production and Animal Health to solicit informed consent to participate in the study.

## 3. Results

A total of 486 dogs from 224 households surrounding 44 GPS points in six districts of Hanoi were examined for the presence of antibodies against flaviviruses (Table 1). The geographical distribution of dogs tested in individual households is shown in Figure 1. 

### 3.1. Seroprevalence by cELISA

The cELISA results revealed 336 positive samples to the pr-E protein of WNV, 139 negative samples and 11 doubtful samples. The doubtful samples were removed from further analyses.

Out of the remaining 475 dogs in 221 households, the flavivirus seropositivity was 70.7% (95% CI = 66.4–74.8%). The seroprevalences of males and females were 73.2% (95% CI = 66.6–79.0%) and 75.7% (95% CI = 67.9–82.1%), respectively. 

Seroprevalences of crossbreed (91.3%; 95% CI = 84.1–95.5%) and local breed (71.6%; 95% CI = 65.6–77.7%) were significantly higher than exotic breed (33.3%; 95% CI = 17.3–54.4%). Significantly higher seropositivity was found in dogs under 12 months old (77.5%; 95% CI = 72.1–82.1%) compared to dogs above 12 months old (56.1%; 95% CI = 46.5–65.2%). The seroprevalence was significantly higher in rural districts (93.3%; 95% CI = 88.1–96.4%) as compared to peri-urban districts (74.5%; 95% CI = 68.5–79.7%) and urban districts (22.5%; 95% CI = 14.9–32.4%). Dogs kept outside the house showed a significantly higher seroprevalence (79.5%; 95% CI = 70.9–86.0%) than indoor dogs (58.2%; 95% CI = 47.8–68.0%). 

Univariable analyses at the animal level (Table 2) identified significant associations between seropositive dogs and district location, age of dog, breed of dog and keeping practice of dog in the households.

### 3.2. Univariable Analysis Results at Household Level

The results obtained by univariable analyses at the household level (Table 3) revealed that households with seropositivity against flavivirus were significantly associated with district location, presence of livestock such as pigs or chickens, mosquito-borne disease history in the family and mosquito coil burning measures. 

On the one hand, the risk of households being seropositive in rural districts (OR = 40.6, *p* < 0.001) and peri-urban districts (OR = 12.8, *p* < 0.001) was significantly higher than in urban districts. Likewise, the risk of seropositivity was higher (*p* < 0.01) in households keeping livestock, pigs and/or chickens than houses without livestock. Families without a reported previous human case of a mosquito-borne disease had a higher risk of having seropositive dogs (OR = 5.96, *p* = 0.002).

There was no significant difference in seroprevalence depending on the presence of cats in the houses.

On the other hand, burning coils to control mosquitoes was significantly associated with an increased proportion of positive households (OR = 3.263, *p* = 0.019). Other practices at households including door/window screening, use of repellent, mosquito net, mosquito electric trap or racket, lid covered on water tanks, larvicides, insecticides, eliminating the breeding site of mosquitoes and keeping fish in water tanks did not show a significant risk associated with flavivirus exposure.

### 3.3. Multivariable Analysis Results

The paired variables between district location and livestock keeping (r = −0.75), between district location and pig (r = −0.71) and between livestock keeping and pig production (r = 0.89) were strongly correlated. Of the 81 households keeping livestock, 86% (n = 70) of the households kept pigs and the final models of pig keeping and livestock keeping variables were not different. Therefore, the variable for livestock keeping was taken out from the modeling. There were two multivariable logistic regression models built (Table 4).

In model 1 that excluded the variables of livestock keeping and pig keeping, coil burning had an effect of more than 33% on the coefficient of larvicides use and the change in this exposure variable became insignificant; therefore, this confounding factor was added back to the model. The final model determined district location and use of larvicides in water tanks were significantly associated with the positivity of houses (*p* < 0.05). 

Model 2 without the variables of district location and livestock keeping identified significant risks of the positivity of households as pig keeping, mosquito electric trap use, coil burning and mosquito-borne disease history of family (*p* < 0.05). 

Both models showed a good fit (Hosmer–Lemeshow statistic test, *p* = 0.114 and 0.541, respectively).

## 4. Discussion

This study indicated that the overall seroprevalence against flaviviruses in dogs in Hanoi was as high as 70.7%. By the same cELISA as used here, previously reported prevalences of flavivirus seropositivity in dogs have been highly varied, e.g., 5.7% in China [8], 62% in Morocco [16] and 42.1% in Romania [17]. Notwithstanding the circulation in Cambodia, Myanmar, Thailand, Indonesia, Malaysia, the Philippines and China, WNV has never been reported in Vietnam [18]. However, several other flaviviruses have been long present in Vietnam. Specifically, JEV was isolated already in 1951 and subsequent virus isolations have been performed from humans, pigs and birds [19,20]. In 1958, DENV was first reported in the northern region of Vietnam [21]; seropositivity for Zika was first confirmed in 1954 and since then, the first cases of ZIKV were reported in 2015 [22]. Flaviviruses are well known to have serological cross-reactions, and therefore the results of the ELISA are not sufficiently virus-specific. The plaque reduction neutralization test (PRNT), the most specific serological test for flaviviruses [23], was not conducted against all flaviviruses endemically circulating in Vietnam as well as WNV; therefore, we acknowledge this limitation in our study. Serum neutralization assays for multiple flaviviruses are suggested for confirmation in future studies. 

Dog puppies under 12 months of age showed more than two times higher odds of having flavivirus antibodies as compared to adult dogs. Naturally, old animals have had more chances of being exposed to infectious agents. However, maternal antibodies from infected mother dogs to their puppies through colostrum could be maintained for a period, which could explain this higher rate. However, maternal immunity to flaviviruses in dogs is still unknown. Generally, immunity relies on flavivirus antibody persistence in vertebrate hosts, but the mechanism(s) for persistence is still poorly understood [24,25]. Further studies on flavivirus immunology post-infection in dogs are suggested. 

Dogs with greater outdoor exposure obtained a higher level of flavivirus seroprevalence, which was also consistent with earlier findings [8,26]. Generally, exotic breeds of dogs imported to Vietnam have a very high economical value and they are closer to their owner. In contrast, crossbreed and indigenous dogs that have lower value may be kept outdoors more; therefore, they have a greater possibility of being infected with flaviviruses through mosquito feeding. Significantly higher seroprevalences to flavivirus of local and crossbreed dogs due to more frequent outdoor keeping than exotic ones were also found in this study. 

Location of households was significantly associated with seropositivity among the dogs, while sex of dogs was not a risk factor, which is similar to a previous study [8]. Rural dogs and peri-urban dogs showed forty times and eighteen times higher risk of exposure as compared to urban dogs, respectively. This is likely related to the greater livestock presence in rural than in urban Hanoi [5], since more mosquitoes have been found in livestock shelters than in non-livestock-keeping households [27]. In fact, vector distribution varies depending on mosquito species. For instance, *Culex* spp. mosquitoes, the major vector of JEV, prefer to breed in polluted aquatic habitats such as rice production areas, wetlands and ponds [28]. A previous study found high abundance of *Culex tritaeniorhynchus, Cx. vishnui, Cx. gelidus* and *Cx. fuscocephala* in cultivating rice fields of a Hanoi rural area [29]. In contrast, *Aedes* spp. mosquitoes, the main vector for dengue, have been more adapted to human environments and have their breeding sites in clean and undisturbed water; therefore, they are close to aquatic items surrounding human dwellings [30]. An entomological survey in Hanoi revealed that concrete water tanks, clay jars and drums were abundant in *Aedes aegypti* and *Ae. albopictus* larvae [31].

Households keeping livestock, especially pigs or/and chickens, were at higher risk of flavivirus exposure in dogs than houses without a livestock presence. In Vietnam, the seroprevalence of JEV in pigs, another domestic species, has been reported at above 70% [9,32,33]. Pigs are known amplifying hosts for JEV and known to attract mosquitoes; thus, pig keeping increases the risk of both viral circulation and the number of mosquitoes that act as vectors [34]. 

The risk of flavivirus exposure in dogs was about six times higher in families without historical infection of mosquito-borne diseases compared to households that reported human cases of DENV or JEV. The reduced risk of flavivirus seropositivity in dogs in households with experience of mosquito-borne disease could also be correlated to the risk mitigation behavior of these families. 

A limitation of our study is that mosquito control practices of households were not directly observed, or the frequency of implementation was not recorded; therefore, the evaluation of mosquito prevention could not be performed.

However, some mosquito prevention measures applied in households were reported in this study. In particular, window screening can block entry points for mosquitoes in a house [35], while repellents can influence mosquito olfaction [36]. Mosquito bed nets are the most commonly used measure by people in Hanoi, followed by anti-mosquito products such as insecticides, elimination of breeding sites and electric rackets, in order to prevent mosquitoes [37]. Water containers are the most likely breeding sites of some mosquito species such as *Ae. albopictus*, *Ae. niveus* gp. and *Cx. quinquefasciatus* [38]. Previous studies in Vietnam concluded that the use of an appropriate cover on water storage containers effectively reduces pre-adult mosquito infestation levels [39,40]. Keeping fish as predators of mosquito larvae, a biological control for mosquito-borne diseases, has also been studied [41,42].

However, some methods that protect humans and pets from mosquito bites in this study such as using a window screen, repellents, mosquito nets and insecticides spraying did not show any efficiency of lowering seropositivity against flavivirus for the dogs. 

Burning a mosquito coil indoors generates smoke that can control mosquitoes [43,44,45]. In this study, a significant increase in the seropositivity of dogs in the houses burning coils as compared to dogs from the houses without coils was identified, but dogs’ behavior in reaction to coil smoke is unclear. If dogs are sensitive to coil smoke, they may avoid it the same way as mosquito vectors and thus still be at risk of infection. A previous study suggested that mosquito coils do not significantly affect the risk of a mosquito-borne disease in humans if the coils are burnt just once per week [46]. 

Multivariable logistic regression models identified more rural location, pig grazing, no application of mosquito control measures such as electric rackets or larvicides in home water tanks, burning coils and family without mosquito-borne disease experience as the main risk factors for flavivirus exposure of dog-keeping households.

## 5. Conclusions

This study indicated a high flavivirus seroprevalence in the dog population of Hanoi. The main risk factors for households were rural area, presence of pigs, coil burning, no use of either larvicides or mosquito electric racquet/trap at home and no experience of mosquito-borne infections.

Some common mosquito control measures of local people in Hanoi did not significantly mitigate the risk of flavivirus infection in the households.

Understanding the risk factors associated with flavivirus prevalence in dogs could facilitate better mosquito-borne disease control in Hanoi.

## Figures and Tables

**Figure 1 viruses-13-00507-f001:**
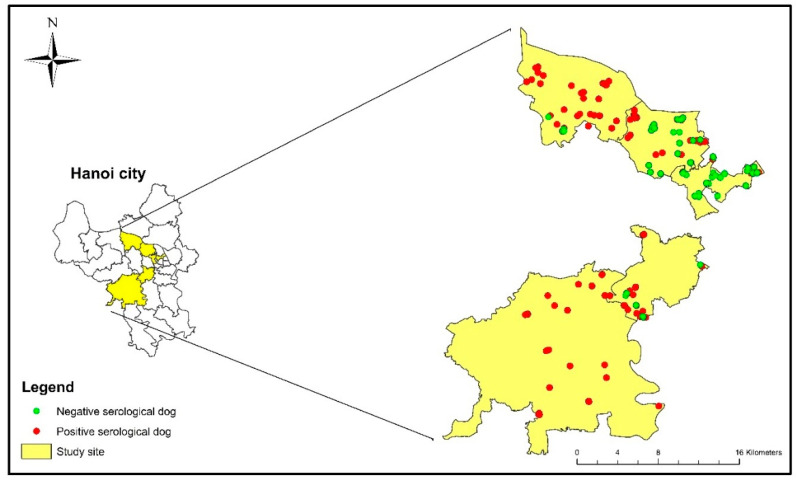
Distribution of flavivirus seroprevalence in dogs rising in Hanoi by cELISA.

**Table 1 viruses-13-00507-t001:** Number of households visited with dogs sampled.

Category	Number of Dogs Sampled	Number of Households
Not enough serum	16	1
cELISA doubtful	11	3
cELISA positive	336	221
cELISA negative	139	
Sum	502	225

**Table 2 viruses-13-00507-t002:** Results from univariable analysis showing the association between seropositivity of dogs and exposure variables.

Exposure Variable	Label	Total Test	Positive	Seroprevalence (95% CI)	OR (95% CI)	*p*-Value
Sex	Male	198	145	73.2 (66.6–79.0)	0.88 (0.52–1.48)	0.607
	Female	140	106	75.7 (67.9–82.1)	1	
Breed	Local	215	153	71.6 (65.6–77.7)	4.94 (1.87–13.9)	<0.001
	Crossbreed	104	95	91.3 (84.1–95.5)	21.11 (6.28–72.6)	
	Exotic	24	8	33.3 (17.3–54.4)	1	
Age group	≤12 months	271	210	77.5 (72.1–82.1)	2.70 (1.62–4.46)	<0.001
	>12 months	107	60	56.1 (46.5–65.2)	1	
District	Rural	151	141	93.3 (88.1–96.4)	48.6 (20.4–121)	<0.001
	Peri-urban	235	175	74.5 (68.5–79.7)	10.06 (5.47–18.9)	
	Urban	89	20	22.5 (14.9–32.4)	1	
Dog keeping at house	Outside	112	89	79.5 (70.9–86.0)	2.77 (1.43–5.42)	0.001
	Inside	91	53	58.2 (47.8–68.0)	1	

Abbreviations: OR, odds ratio; CI, confidence interval.

**Table 3 viruses-13-00507-t003:** Results from univariable analysis showing the association between seropositivity of households and exposure variables.

Exposure Variable	Label	Total HH Tested	HH Positive	OR (95%CI)	*p*-Value
District	Rural	71	67	40.6 (12.9–127)	<0.001
	Peri-urban	85	75	18.2 (7.77–42.5)	<0.001
	Urban	65	19	1	-
Household keeping livestock in general	Yes	81	75	8.13 (3.31–20.0)	<0.001
	No	137	83	1	
Household keeping pig	Yes	70	64	6.13 (2.49–15.1)	<0.001
	No	148	94	1	
Household keeping chicken	Yes	38	35	5.41 (1.60–18.3)	0.007
	No	180	123	1	
Household that has cat	Yes	32	27	2.27 (0.83–6.19)	0.110
	No	186	131	1	
Family member no experience with mosquito disease	Yes	199	149	5.96 (1.94–18.3)	0.002
	No	15	5	1	
**Mosquito Prevention Practice by Using:**					
Window/door screen	Yes	23	15	0.71 (0.28–1.76)	0.457
	No	190	138	1	
Repellent	Yes	39	31	1.65 (0.71–3.83)	0.243
	No	174	122	1	
Mosquito net	Yes	189	137	1.32 (0.53–3.26)	0.551
	No	24	16	1	
Electric racket/portable electric trap	Yes	126	86	0.64 (0.34–1.20)	0.164
	No	87	67	1	
Mosquito coil/incense stick	Yes	40	35	3.26 (1.21–8.78)	0.019
	No	173	118	1	
Lid covered on water tank	Yes	77	50	0.59 (0.32–1.09)	0.094
	No	136	103	1	
Chemical/larvicide in water container	Yes	12	7	0.527 (0.16–1.73)	0.291
	No	201	146	1	
Insecticides spraying	Yes	102	74	1.07 (0.59–1.95)	0.823
	No	111	79	1	
Breeding site elimination	Yes	51	40	1.58 (0.75–3.33)	0.232
	No	162	113	1	
Fish in water container	Yes	69	54	1.64 (0.84–3.20)	0.151
	No	144	99	1	

Abbreviations: HH, household; OR, odds ratio; CI, confidence interval.

**Table 4 viruses-13-00507-t004:** Final multivariable analysis of risk factors for dog-keeping households.

Exposure Variable	Categories	Coef.	ORs	95% CI	*p*-Value
**Model 1. Without the variables for livestock keeping and pig keeping**					
District	Rural	3.70	40.6	12.3–134	<0.001
	Peri-urban	2.81	16.7	6.96–40.2	<0.001
	Urban	Ref	Ref		
No larvicides in water containers	Yes	1.68	5.39	1.06–27.3	0.042
	No	Ref	Ref		
Coil burning	Yes	0.78	2.18	0.58–8.11	0.247
	No	Ref	Ref		
**Model 2. Without the variables for district location and livestock keeping**					
Pig keeping	Yes	1.75	5.76	2.27–14.6	<0.001
	No	Ref	Ref		
No use of mosquito electric racket/trap	Yes	0.73	2.08	1.05–4.14	0.036
	No	Ref	Ref		
Coil burning	Yes	1.13	3.09	1.04–9.17	0.042
	No	Ref	Ref		
No experience with mosquito disease in family	Yes	1.60	4.94	1.50–16.3	0.009
	No	Ref	Ref		

Abbreviations: Coef., coefficients; Ref, reference; OR, odds ratio; CI, confidence interval.

## Data Availability

All datasets supporting our findings are available from the corresponding author on reasonable request.

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
