# Peer review of "Dogs as Sentinels for Flavivirus Exposure in Urban, Peri-Urban and Rural Hanoi, Vietnam"

_viruses, 2021, doi:10.3390/v13030507_

Round 1

Reviewer 1 Report

Dogs as sentinels for flavivirus exposure in urban, peri-urban and rural Hanoi, Vietnam

By Pham-Thanh et al.

The author presents a straight forward study. However, I think that the manuscript will benefit from some changes:

  • The title of the article is proper since the ELISA that was used in this study is general to flavivirus. However, in the end, the study shows high exposure of dogs to Japanese encephalitis virus by using specific VNT. Since the authors did not use specific VNT’s to other flaviviruses, I think that emphasis on Japanese encephalitis virus should be mentioned more. Specifically, in the objectives of this study but it should also be mentioned more in the discussion and conclusions.
  • In the last part of the results and discussion, there is a lot of epidemiological data that was collected, presented and conclusions are drawn accordingly. However, there is no mention of how these epidemiological data were collected and investigated in the material and methods. Such a section needs to be added.

Author Response

Point 1: The title of the article is proper since the ELISA that was used in this study is general to flavivirus. However, in the end, the study shows high exposure of dogs to Japanese encephalitis virus by using specific VNT. Since the authors did not use specific VNT’s to other flaviviruses, I think that emphasis on Japanese encephalitis virus should be mentioned more. Specifically, in the objectives of this study but it should also be mentioned more in the discussion and conclusions.

Response 1: A major limitation of this study is that we could not confirm all samples by VNT for all potential flaviviruses that could be circulating in Hanoi and WNV. There a few samples were tested by JEV PRNT that did not provide enough evidence to draw a conclusion for JEV seropositive representation of dog population in Hanoi. Therefore, we removed JEV PRNT results in this analysis to make the analysis more specific and consistent.

Point 2: In the last part of the results and discussion, there is a lot of epidemiological data that was collected, presented and conclusions are drawn accordingly. However, there is no mention of how these epidemiological data were collected and investigated in the material and methods. Such a section needs to be added.

Response 2Thank you for this comment and we provided more information relating to epidemiological study design in the section for materials and methods of this attached revised manuscript version.

We would like to submit the revised manuscript and the additional texts are highlighted in red color.

Reviewer 2 Report

In the study, the authors screened IgG against Pan-Flaviviruses by competitive ELISA kit for West Nile virus, using 475 dog sera in Hanoi, Vietnam. PRNT for JEV was conducted in randomly selected 34 positive and 10 negative samples setting the cut off of 1:40. Additionally the authors collected several epidemiological data such as dog breed, location of house holder and mosquito prevention practice. The authors concluded that the dog can serve as a sentinel of JEV in the region, but the data is not enough to conclude for that.

Major points

The authors develop their discussion about virus species (DENV, JEV, WNV and ZIKV) based on their results of PRNT, but they only evaluated JEV. To confirm that the majority of the dog infection is derived from JEV, the neutralization activity should be compared with other possible Flaviviruses i.e. DENV, WNV and ZIKV.

Both JEV and WNV belong same JEV serogroup and their seroreactivity is similar, showing some sero-cross-reaction. As referred in the manuscript, WNV cases have been reported in Southeast Asian countries around Vietnam. To confirm the specificity of neutralization activity against JEV, I highly recommend to compare with WNV using the same 34 positive and 10 negative samples.

In the discussion, the authors focused on several Flaviviruses. Each virus are very different from each other in the point of their host, transmission cycle, and vector. Especially the biological behavior of their vectors, Aedes spp. and Culex spp. vary and it may affect the results of epidemiological analysis, because their vectorial capacities of each Flaviviruses are different too. But the manuscript lacks the discussion about vector character.

Minor points

Abstract

“These results will support better control of mosquito-borne diseases in Hanoi and indicated that dogs can be used as sentinels for flaviviruses exposure, especially JEV.”

The data is not enough to conclude that because the possibility of cross-reaction can’t be excluded.

Introduction

“During 2017 and 2018, respectively, 38,131 and 3,694 human cases of dengue fever were reported in Hanoi [7], [8].”

Please show the number of JEV patient because the topic of this manuscript is mainly about JEV.

Materials and Methods

“A total of six districts including two rural, two peri-urban and two urbans in Hanoi were purposively selected to represent a gradient of livestock keeping.”

What is the definition of rural, peri-urban and urban? Please indicate the evidence of each categories, such as population density.

“A subset of randomly selected serum samples including 34 positive and 10 negative sera as determined by cELISA was analyzed by plaque reduction neutralization test (PRNT).”

How did authors select the 44 samples? 14 or 15 samples from rural, peri-urban and urban area? The collection site affect the result of serological test. Generally speaking Aedes aegypti prefer human living environment and Culex tritaeniorhyncus prefer large area such as rice field, transmitting DENV and JEV respectively. The subpopulation should reflect the target population.

Discussion

“most of the infections in dogs were due to JEV,”

As written above, the data is not enough to conclude for that.

“Inappropriate practice for mosquito control of the householders would be a reason.”

I can’t find the evidence of above conclusion in the manuscript.

Author Response

Point 1. In the study, the authors screened IgG against Pan-Flaviviruses by competitive ELISA kit for West Nile virus, using 475 dog sera in Hanoi, Vietnam. PRNT for JEV was conducted in randomly selected 34 positive and 10 negative samples setting the cut off of 1:40. Additionally the authors collected several epidemiological data such as dog breed, location of house holder and mosquito prevention practice. The authors concluded that the dog can serve as a sentinel of JEV in the region, but the data is not enough to conclude for that. 

Response 1. Thank you. We have revised this to be more general for flavivirus.

Major points

Point 2. The authors develop their discussion about virus species (DENV, JEV, WNV and ZIKV) based on their results of PRNT, but they only evaluated JEV. To confirm that the majority of the dog infection is derived from JEV, the neutralization activity should be compared with other possible Flaviviruses i.e. DENV, WNV and ZIKV.

Response 2. Lack of PRNT for different flaviviruses is a major limitation of the study and we acknowledge this constraint in the manuscript. Since a small group of dog sera was tested for JEV by PRNT, that is not representative for whole blood samples, we have decided to remove JEV PRNT from the analysis.

Point 3. Both JEV and WNV belong same JEV serogroup and their seroreactivity is similar, showing some sero-cross-reaction. As referred in the manuscript, WNV cases have been reported in Southeast Asian countries around Vietnam. To confirm the specificity of neutralization activity against JEV, I highly recommend to compare with WNV using the same 34 positive and 10 negative samples.

Response 3. Thank you very much for this important point. However, no lab capacity for WNV diagnosis especially PRNT in Vietnam is available at present (for instance BSL-3, WNV seed for neutralization).

Point 4. In the discussion, the authors focused on several Flaviviruses. Each virus are very different from each other in the point of their host, transmission cycle, and vector. Especially the biological behavior of their vectors, Aedes spp. and Culex spp. vary and it may affect the results of epidemiological analysis, because their vectorial capacities of each Flaviviruses are different too. But the manuscript lacks the discussion about vector character.

Response 4. Thank you for this great comment. We have added more discussions on this topic.

Minor points

Point 5. Abstract: “These results will support better control of mosquito-borne diseases in Hanoi and indicated that dogs can be used as sentinels for flaviviruses exposure, especially JEV.”

The data is not enough to conclude that because the possibility of cross-reaction can’t be excluded.

Response 5. Yes, it is so we removed JEV implication in the abstract

Point 6. Introduction: “During 2017 and 2018, respectively, 38,131 and 3,694 human cases of dengue fever were reported in Hanoi [7], [8].”

Please show the number of JEV patient because the topic of this manuscript is mainly about JEV.

Response 6. Yes, we agreed, so we replaced this information with another citation from our published paper that concluded DEN and JEV are major flaviviruses in the north of Vietnam, including Hanoi.

 Point 7.  Materials and Methods

“A total of six districts including two rural, two peri-urban and two urbans in Hanoi were purposively selected to represent a gradient of livestock keeping.”

What is the definition of rural, peri-urban and urban? Please indicate the evidence of each categories, such as population density.

Response 7. We have provided a definition of rural, peri-urban and urban based on livestock population of Hanoi in the revised manuscript.

 Point 8.

“A subset of randomly selected serum samples including 34 positive and 10 negative sera as determined by cELISA was analyzed by plaque reduction neutralization test (PRNT).”

How did authors select the 44 samples? 14 or 15 samples from rural, peri-urban and urban area? The collection site affect the result of serological test. Generally speaking Aedes aegypti prefer human living environment and Culex tritaeniorhyncus prefer large area such as rice field, transmitting DENV and JEV respectively. The subpopulation should reflect the target population.

Response 8. We have removed this content from the manuscript to make more specific analysis on general flaviviruses

Point 9. Discussion

“most of the infections in dogs were due to JEV,”

As written above, the data is not enough to conclude for that.

Response 9. Yes, we agreed, and we corrected this content

 Point 10.

“Inappropriate practice for mosquito control of the householders would be a reason.”

I can’t find the evidence of above conclusion in the manuscript.

Response 10. Since we could not provide evidence for this comment because no mosquito control evaluation was undertaken in the study, we also acknowledge this limitation, we removed this sentence from the manuscript.

We would like to submit the revised manusctipt with additional information in red texts.

Round 2

Reviewer 2 Report

The results of PRNT should be shown as supplemental data. Even though the virus species (JEV) and the number of sample are limited, the information is helpful for better understanding of readers and support the discussion.

Additionally I recommend that the authors show the format of questionnaires sheet as supplemental data (not the results, but the format).

Line 246-247

Cx. vishnui, Cx. gelidus

Line 269

Please indicate the species name of mosquitoes.

Line 276-277

According to the previous version of the manuscript, the authors supposed most dog infections are derived from JEV. And as written in the current manuscript, Culex, a major JEV vector prefers rice product area, wetlands and ponds. Consequently even if “lid covered on water tank”, “mosquito breeding site elimination” and “fish keeping in water containers of the households” did not show any efficiency of lowering seropositivity against flavivirus for the dogs, it's not to be wondered at. The above three categories should be removed from the sentence.

Line 290-295

These sentences should be moved to the appropriate paragraph in the discussion, but not appear in the last paragraph.

Line 304

Still I think the data is not enough to conclude for that. The conclusion does not meet to the title and the results of the manuscript.

Author Response

Point 1. The results of PRNT should be shown as supplemental data. Even though the virus species (JEV) and the number of samples are limited, the information is helpful for better understanding of readers and support the discussion.

Response 1. We agreed on the suggestion and the results of PRNT for JEV are shown in a table as supplemental data.

Point 2. Additionally I recommend that the authors show the format of questionnaires sheet as supplemental data (not the results, but the format).

Response 2. The format of questionnaire form that we used to collect data in the field is also shown as supplemental data

Point 3. Line 246-247

Cx. vishnuiCx. gelidus

Response 3. We thank you for your correction, the names of two species were corrected.

Point 4. Line 269

Please indicate the species name of mosquitoes.

Response 4. We have added the names of the species of Ae. albopictus, Ae. niveus gp. and Cx. quinquefasciatus that their preferable breeding sites are several water container types in the manuscript.

Point 5. Line 276-277

According to the previous version of the manuscript, the authors supposed most dog infections are derived from JEV. And as written in the current manuscript, Culex, a major JEV vector prefers rice product area, wetlands and ponds. Consequently even if “lid covered on water tank”, “mosquito breeding site elimination” and “fish keeping in water containers of the households” did not show any efficiency of lowering seropositivity against flavivirus for the dogs, it's not to be wondered at. The above three categories should be removed from the sentence.

Response 5. We are convinced with this comment and the three categories have been removed.

Point 6. Line 290-295

These sentences should be moved to the appropriate paragraph in the discussion, but not appear in the last paragraph.

Response 6. We have moved the sentences on limitations in the study in other paragraphs that are relevant to them.

Point 7. Line 304

Still I think the data is not enough to conclude for that. The conclusion does not meet to the title and the results of the manuscript.

Response 7. We are convinced with your comment and we have removed the sentence “Our study strongly supported earlier evidence that dogs can serve as efficient sentinels for JEV-exposure to humans” from our conclusion.

Our changings were highlighted with "red sentences" in the revised manuscript